# Learn what matters: cross-domain imitation learning with task-relevant embeddings

**Tim Franzmeyer**
University of Oxford
frtim@robots.ox.ac.uk

**Philip H. S. Torr**
University of Oxford
philip.torr@eng.ox.ac.uk

**João F. Henriques**
University of Oxford
joao@robots.ox.ac.uk

## Abstract

We study how an autonomous agent learns to perform a task from demonstrations in a different domain, such as a different environment or different agent. Such cross-domain imitation learning is required to, for example, train an artificial agent from demonstrations of a human expert. We propose a scalable framework that enables cross-domain imitation learning without access to additional demonstrations or further domain knowledge. We jointly train the learner agent's policy and learn a mapping between the learner and expert domains with adversarial training. We effect this by using a mutual information criterion to find an embedding of the expert's state space that contains task-relevant information and is invariant to domain specifics. This step significantly simplifies estimating the mapping between the learner and expert domains and hence facilitates end-to-end learning. We demonstrate successful transfer of policies between considerably different domains, without extra supervision such as additional demonstrations, and in situations where other methods fail.

## 1 Introduction

Reinforcement learning (RL) has shown great success in diverse tasks and distinct domains [43, 2], however its performance hinges on defining precise reward functions. While rewards are straightforward to define in simple scenarios such as games and simulations, real-world scenarios are significantly more nuanced, especially when they involve interacting with humans.

One possibility for overcoming the problem of reward misspecification is to learn policies from observations of expert behaviour, also known as imitation learning. Recent imitation learning algorithms rely on updating the learner agent's policy until the state occupancy of the learner matches that of the expert demonstrator [4], requiring the learner and expert to be in the same domain. Such a requirement rarely holds true in more realistic scenarios. Consider for example the case where a robot arm learns to move an apple onto a plate from demonstrations of a human performing this task. Here, both domains do inherently share structure (the apples and the plates have similar appearances) but are distinct (the morphologies, dynamics and appearances of the two arms are different).

Enabling a learner agent to successfully perform a task from demonstrations that were generated by a different expert agent, which we refer to as a different domain even if the tasks are related, would widely broaden the possibilities to train artificial agents. This cross-domain imitation learning problem is seen as an important step towards value alignment, as it facilitates transferring behaviour from humans to artificial agents [32, Chapter 7].

This problem has only been considered by researchers in realistic settings recently. Due to its difficulty, previous work on cross-domain imitation learning either assumes the expert's and learner's domains to be almost identical [42, 17, 6], requires demonstrations of experts in multiple domains that are similar to the learner's [45, 44], or relies on the availability of demonstrations of proxy

36th Conference on Neural Information Processing Systems (NeurIPS 2022).

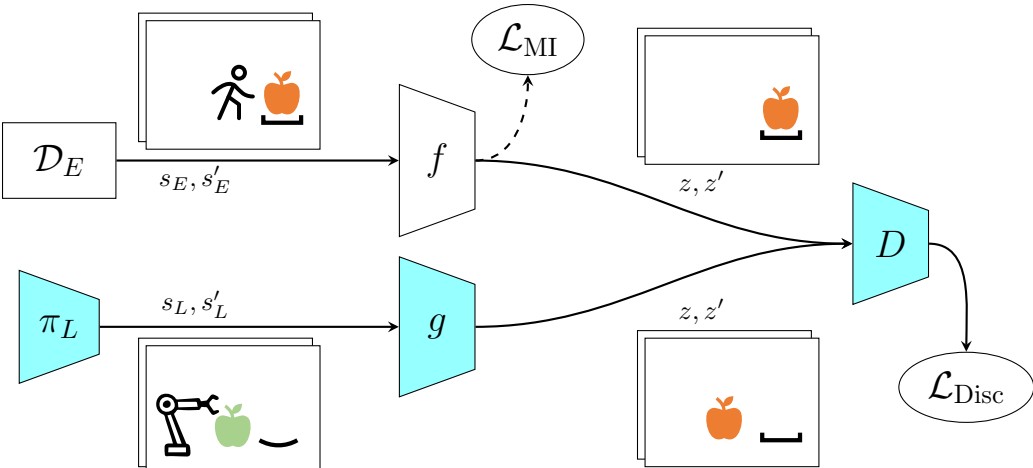

Figure 1: We consider a robot learning to place an apple onto a plate from demonstrations of a human doing so. This illustrative cross-domain imitation learning problem requires finding the learner's policy $\pi_L$ in its domain with states $s_L$ from demonstrations generated by the human expert ($\mathcal{D}_E$) in the distinct expert domain with states $s_E$. We first use a mutual information criterion ($\mathcal{L}_{\text{MI}}$) to find an embedding function $f$ that maps the expert state $s_E$ to a task-relevant representation $z$ to discard domain specific information. In the given example, $f$ would primarily encode information about the apple and the plate, as these are most relevant to the task. We next apply an adversarial loss $\mathcal{L}_{\text{Disc}}$ to jointly train all blue-shaded components, i.e., the policy of the learner ($\pi_L$), the discriminator $D$ and the mapping function f which maps the learner states to the task-relevant representation $z$ of the expert domain. Here, the learner encoder maps the apple's color and the type of plate to that of the expert domain.

tasks in both domains [30, 18]. Designing such proxy tasks is a manual process that requires prior knowledge about both domains, since they have to be inherently similar to the target task to convey a relevant mapping between domains [18]. Fickinger et al. [10] overcome the need for proxy tasks by directly comparing distributions in both domains, effectively addressing the same problem setting as us. While very promising, its applicability is limited to short demonstrations and Euclidean spaces.

We propose to jointly learn the learner policy and the mapping between the learner and expert state spaces, utilizing adversarial training. Unlike standard generative adversarial imitation learning [16, 39], we use domain-specific encoders for both the learner and expert. We therefore devise a mutual information criterion to find an expert encoder that preserves task-relevant information while discarding domain specifics irrelevant to the task. Note that in general, cross-domain imitation learning is an under-defined problem, as a unique optimal policy for the learner is not defined as part of the problem: for example, should a humanoid agent that imitates a cheetah crawl (imitating its gait) or walk (moving in the same direction)?

We evaluate our cross-domain imitation learning approach in different cross-embodiment imitation learning scenarios, comparing on relevant benchmarks, and find that our method robustly learns policies that clearly outperform the baselines. We conduct several ablation studies, in particular finding that we can control how much domain-specific information is transferred from the expert—effectively interpolating between mimicking the expert's behaviour as much as possible and finding novel policies that use different strategies to maximize the expert's reward.

Our contributions are:

- We propose a mutual information criterion to find an embedding of the expert state which contains task-relevant information, while discarding domain specifics irrelevant to the task.
- We learn the mapping between the learner domain and the task-relevant embedding without additional proxy task demonstrations.
- We demonstrate training robust policies across diverse environments, and the ability to modulate how information flows between the learner and expert domains.

## 2 Related Work

**Imitation learning** considers the problem of finding an optimal policy for a learner agent from demonstrations generated by an expert agent, where inverse reinforcement learning (IRL) [1, 46] recovers a reward function under which the observed expert's behaviour is optimal. More recent works [16, 11, 39] define imitation learning as a distribution matching problem and use adversarial training [14] to directly find the learner's policy, without explicitly recovering the expert's reward.

**Cross-domain imitation learning** generalizes imitation learning to the case where the learner and expert are in different domains. Small mismaches between the domains, such as changes in viewpoint or gravitational force, or small variations of the dynamics, are addressed by [42, 12, 17, 28, 36, 8] and Bohez et al. [6]. To learn policies cross-domain in the presence of larger mismatches, such as different embodiments of the learner and the expert, previous works used demonstrations of proxy tasks to learn a mapping between the learner and expert domain, which is then used to find the learner's optimal policy [15, 23, 35, 30, 18], utilized a latent embedding of the environment state [45, 44], or assumed the reward signal to be given [34]. GWIL [10] does not rely on proxy tasks and minimizes the distance between the state-action probability distributions of both agents which lie in different spaces [25]. This approach assumes Euclidean spaces and is computationally intractable when using longer demonstrations, which generally improve the performance of learning algorithms when available.

Our approach obviates the need for proxy tasks, scales to detailed demonstrations of complex behaviours, and enables the control of how much domain-specific information is transferred to the learner domain.

In classical RL [26], where behaviour is learned from a given reward function, **mutual information objectives** are commonly used to find compact state representations that increase performance by discarding irrelevant information [29, 3, 37, 24, 22]. We propose to similarly learn a representation of the expert's state that contains task-relevant information while being invariant to domain specifics.

## 3 Background

**Definitions.** Following Kim et al. [18], we define a domain as a tuple $(\mathcal{S}, \mathcal{A}, \mathcal{P}, \zeta)$, where $\mathcal{S}$ denotes the state space, $\mathcal{A}$ is the action space, $\mathcal{P}$ is the transition function, and $\zeta$ is the initial distribution over states. Given an action $a \in \mathcal{A}$, the distribution over the next state is given by the transition function as $\mathcal{P}(s'|s, a)$. An infinite horizon Markov decision process (MDP) is defined by adding a reward function $r : \mathcal{S} \times \mathcal{A} \to \mathbb{R}$, which describes a specific task, and a discount factor $\gamma \in [0, 1]$ to the domain tuple. We define the expert agent's MDP as $\mathcal{M}_E = (\mathcal{S}_E, \mathcal{A}_E, \mathcal{P}_E, r_E, \gamma_E, \zeta_E)$, and its policy as a map $\pi_E : \mathcal{S}_E \to \mathcal{B}(\mathcal{A}_E)$, where $\mathcal{B}$ is the set of all probability measures on $A_E$. We define the learner MDP $\mathcal{M}_L$ and learner policy $\pi_L$ analogously, except that the learner MDP has no reward function or discount factor. An expert trajectory is a sequence of states $\tau_E = \{s_E^0, s_E^1, \ldots, s_E^n\}$, where $n$ denotes the length of the trajectory. We denote $\mathcal{D}_E = \{\tau_i\}$ to be a set of such trajectories.

**Problem Definition.** The objective of cross-domain imitation learning is to find a policy $\pi_L$ that optimally performs a task in the learner domain $\mathcal{M}_L$, given demonstrations $\mathcal{D}_E$ in the expert domain $\mathcal{M}_E$. In contrast to most prior work, we do not assume access to a dataset of proxy tasks—simple primitive skills in both domains that are similar but different from the inference task—to be given. We do not assume access to the expert demonstration's actions, which may be non-trivial to obtain, e.g., when learning from videos or human demonstrations, and therefore consider the expert demonstrations to consist only of states.

**Adversarial Imitation Learning from Observations.** We first consider the equal-domain case in which both MDPs are equivalent, i.e., $\mathcal{M}_L = \mathcal{M}_E$, and assume that the expert agent's optimal policy $\pi_E$ under $r_E$ is known. Torabi et al. [39] define a solution to this problem as an extension of the standard imitation learning problem [16], by minimizing the divergence between the learner's state-transition distribution $\rho_{\pi_L}$ and that of the expert $\rho_{\pi_E}$, as

$$\underset{\pi_L}{\arg\min} -H(\pi_L) + \mathbb{D}_{\text{JS}}\left(\rho_{\pi_L}(s, s') - \rho_{\pi_E}(s, s')\right) = \text{RL} \circ \text{IRL}\left(\pi_E\right), \tag{1}$$

where $\mathbb{D}_{\mathrm{JS}}$ is the Jensen-Shannon divergence and $H(\pi_L)$ is the learner's policy entropy [46]. The state-transition distribution for a policy $\pi$ is defined as

$$\rho_\pi(s_i, s_j) = \sum_a P(s_j|s_i, a)\pi(a|s_i) \sum_{t=0}^{\infty} \gamma^t P(s_t = s_i|\pi). \tag{2}$$

In particular, the expert's state-transition distribution $\rho_{\pi_E}$ is estimated using expert demonstrations $\mathcal{D}_E$. The above objective (eq. 1) can also be derived as the composition of the IRL and RL problems, where $r_E = \mathrm{IRL}(\pi_E)$ denotes the solution to the Inverse Reinforcement Learning problem from policy $\pi_E$ and $\pi_L = \mathrm{RL}(r_E)$ denotes the solution to the RL problem with reward $r_E$.

The IRL component, which recovers the reward function $r : \mathcal{S} \times \mathcal{S} \to \mathbb{R}$ under which the expert's demonstrations are uniquely optimal[1] by finding a reward function that assigns high rewards to the expert policy and low rewards to other policies, is given as $\mathrm{IRL}(\pi_E) = \arg\min_r \left( \max_{\pi_L} \mathbb{E}_{\pi_L}[r(s, s')] - \mathbb{E}_{\pi_E}[r(s, s')] \right)$.

## 4  Unsupervised Imitation Learning Across Domains

We first introduce the cross-domain imitation learning problem before deriving an adversarial learning objective that allows the simultaneous training of the learner's policy and a mapping between the MDPs of the learner and expert. We then demonstrate how the cross-domain imitation learning problem can be significantly simplified be finding an embedding of the expert agent's state space that contains task-relevant information while discarding domain-specific aspects. Lastly, we introduce a time-invariance constraint to prevent degenerate mapping solutions. As our approach does not rely on additional demonstrations from proxy tasks, we refer to it as unsupervised cross-domain imitation learning objective (UDIL).

### 4.1  Cross-domain adversarial imitation learning

We consider the case in which the expert's and agent's MDPs are different, i.e., $\mathcal{M}_L \neq \mathcal{M}_E$, such as when learner and expert are of different embodiments or are in different environments. Kim et al. [18] show that, if there exists an injective mapping $g$ that reduces the learner MDP $\mathcal{M}_L$ to the expert MDP $\mathcal{M}_E$, then a policy $\pi_L$ that is optimal in $\mathcal{M}_L$ is also optimal in the $\mathcal{M}_E$.

Since we do not assume extra supervision from the expert's actions, we define the mapping function between the learner and expert MDPs $g : \mathcal{S}_L \to \mathcal{S}_E$ as a mapping between the respective state spaces. We accordingly define the cross-domain adversarial imitation objective as

$$\arg\min_{\pi_L} -H(\pi_L) + \mathbb{D}_{\mathrm{JS}}(\rho_{\pi_L}(g(s_L), g(s'_L)) - \rho_{\pi_E}(s_E, s'_E)). \tag{3}$$

Applying the mapping $g$ to the learner agent's state allows us to compare the learner's and expert's distributions, even though they are defined over different state-spaces.

### 4.2  Reducing the expert's state dimension

The full state of the expert domain $s_E$ generally contains information that is specific to the task which the expert is demonstrating, defined by the expert's reward function $r_E$, as well as information that is specific to the domain but irrelevant to the task itself. We simplify the cross-domain imitation learning problem by reducing the expert agent's state space to a task-relevant embedding that is invariant to domain specifics.

We assume that the learner state $s$ is multi-dimensional and recall the IRL component of the adversarial imitation problem (eq. 1), which finds the reward function under which the expert's behavior is optimal. We define a second mapping function $f : \mathcal{S}_E \to \mathcal{Z}$, that maps the expert states $s_E \in \mathcal{S}_E$ to lower-dimensional representations $z \in \mathcal{Z}$, with $|\mathcal{Z}| \ll |\mathcal{S}_E|$. When $f$ is chosen as a dimension reduction operation that discards state dimensions of which the reward is independent, we can write the IRL component of eq. 1 as a function of only the embedded representation $z$ (proof in app. 8.1.1),[2]

---

[1]We swap the cost function for the reward function and omit the cost function regularization for simplicity.

[2]We assume that the reward function $r$ is also defined on the embedding space $\mathcal{Z}$, see app. 8.1.1 for details.

as

$$\text{IRL}(\pi_E) = \arg\min_r \left( \max_{\pi_L} \mathbb{E}_{\pi_L}[r(z, z')] - \mathbb{E}_{\pi_E}[r(z, z')] \right). \tag{4}$$

**Simplifying the mapping between learner and expert.** Assuming $f$ to be given, we can further redefine the mapping between learner and expert state as $g : \mathcal{S}_L \to \mathcal{Z}$. That is, the state transformation $g$ no longer has to map the learner state to the full expert state, but only to the task-relevant embedding of the expert state. This not only significantly simplifies the complexity of the mapping function $g$, but also prevents transferring irrelevant domain specifics from the expert to the learner domain. We can then rewrite the cross-domain adversarial imitation objective as

$$\arg\min_{\pi_L, g} -H(\pi_L) + \mathbb{D}_{\text{JS}}(\rho_{\pi_L}(g(s_L), g(s'_L)) - \rho_{\pi_E}(f(s_E), f(s'_E))), \tag{5}$$

which minimizes the distance between the transformed distribution over learner states $s_L$ and the distribution over embedded expert states $z$.

### 4.3 Finding a task-relevant embedding

We now detail how to find a embedding function $f$ from the expert demonstrations $\mathcal{D}_E$. We first assemble a set containing all expert transitions $(s_E, s'_E)$ observed in the trajectories of the demonstration set $\mathcal{D}_E$. We then generate a set of pseudo-random transitions $(s_{\text{rand}}, s'_{\text{rand}})$ by independently sampling two states out of all individual states contained in $\mathcal{D}_E$. We then model all state transitions $(s, s')$ and their corresponding labels $y$, indicating whether it is a random or expert transition, as realizations of a random variable $(S, S', Y)$ on $\mathcal{S}_E \times \mathcal{S}_E \times \{0, 1\}$. Note that any time-invariant embedding $f : \mathcal{S}_E \to \mathcal{Z}$ induces a random variable $(Z, Z', Y)$ on $\mathcal{Z} \times \mathcal{Z} \times \{0, 1\}$ via $(Z, Z') = (f(S), f(S'))$. We then define the mapping $f$ as a mapping that maximizes the mutual information $I$ between the label $Y$ and the embedded state transition $(Z, Z')$, that is,

$$\arg\max_f I((Z, Z'); Y) = \arg\max_f I((f(S), f(S')); Y). \tag{6}$$

Observe that maximizing $I(Z; Y)$ would lead to non-informative representations, as the states contained in the random trajectories are indeed states of the expert trajectory; only *state transitions* $(S, S')$ can distinguish between the two.

### 4.4 Avoiding degenerate solutions

Jointly learning the mapping function $g$ and the learner agent's policy $\pi_L$ may lead to degenerate mappings if $g$ is a function of arbitrary complexity. An overly-expressive $g$ can make the divergence between distributions arbitrarily small, regardless of their common structure, by the universality property of the uniform distribution, i.e., any two distributions can be transformed into each other by leveraging their cumulative density functions (CDFs) and inverse CDFs. We prevent these degenerate solutions with an information asymmetry constraint: we ensure that the mapping $f$ is time-invariant, while the JS-divergence compares distributions across time, i.e., in a time-variant manner. A theoretical analysis is presented in app. 8.1.2.

### 4.5 Unsupervised cross-domain adversarial imitation learning

We finally define the unsupervised cross-domain adversarial imitation learning (UDIL) objective as an adversarial learning problem. We iterate between updating the learner agent's policy $\pi_l$, the mapping $g$ between the learner's and expert's state spaces, and the discriminator $D$. The discriminator's objective is to distinguish between state transitions generated by the learner and state transitions generated by the expert, giving the overall objective

$$\min_{g, \pi_L} \max_\theta \mathbb{E}_{\pi_L}[\log(D_\theta(g(s_L), g(s'_L)))] + \mathbb{E}_{\pi_E}[\log(1 - D_\theta(z, z'))]. \tag{7}$$

## 5 Experiments

**Preliminaries.** We test our approach on two different benchmarks that represent multiple domains and different agents with both environment-based and agent-based tasks. We designed our experiments to answer the following questions.

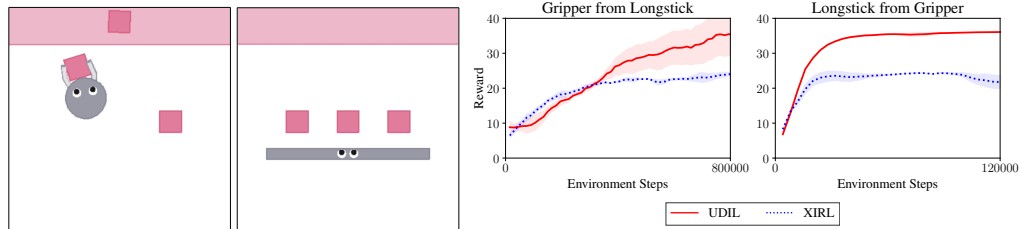

Figure 2: In the XMagical benchmark [40, 45], agents with different embodiments (such as Gripper and Longstick displayed here) have to move the three magenta-colored blocks to the magenta-shaded target zone at the top of the environment. We evaluate the reward achieved by both learner agents when trained on demonstrations of the other using either our algorithm UDIL, or the XIRL [45] baseline.

- Can we find task-relevant embeddings of the expert state *solely* from expert demonstrations, and improve the performance of imitation learning?
- Does the proposed framework robustly learn meaningful policies compared to previous work?
- Can we control the amount of domain-specific information transferred from the expert to the learner?

We compare with the GWIL baseline [10], which is the only other work that makes similar assumptions to ours, i.e., unsupervised cross-domain imitation learning with access only to demonstrations of a single expert agent. In the later presented XMagical environment, we also compare to a modified single-demonstrator-agent version of XIRL [45], which originally relies on demonstrations of multiple distinct expert agents. As no reward function in the learner domain is given, we measure performance of the learner agent by defining its reward as the components of the expert agent's reward function that can be directly transferred to the learner domain. To ensure reproducibility, we run all experiments on random seeds zero to six, report mean and standard error for all experiments (lines and shaded areas), and describe the experiments in full detail in appendix section 8.2.

## 5.1 XIRL baseline

**Setup.** Figure 2 shows the XMagical environment [41, 45] which consists of four agents with different embodiments that have to perform equivalent modifications in the environment, namely pushing all blocks to a shaded region. The corresponding baseline algorithm XIRL [45] trains each agent with demonstrations of the three other expert agents. As our work only requires demonstrations from a single expert agent, we focus on the two most distinct agents, Gripper and Longstick, which are displayed in Figure 2, and evaluate the performance of each when trained on demonstrations of the other. The reward is given as a function of the average distance between the task-relevant objects and their target positions.

**Finding a task-relevant embedding.** The environment state in XMagical is given as a multidimensional vector that describes different absolute and relative positions of environment objects and the agent itself. To find the task-relevant embedding of this state we first generate sets of expert and pseudo-random transitions, as described in section 4.3. As maximizing mutual information objectives in large continuous domains is intractable [5, 9], we instead approximate the objective in eq. (6) by first computing the empirical mutual information between state transitions and labels for each individual state dimension, using the method of Ross [31]. We then find the task-relevant embedding by selecting the dimensions with highest mutual information using the elbow method [19]. We find a clear margin between those state dimensions that are intuitively relevant to the task, such as dimensions that describe the positions of the blocks, and those dimensions that are intuitively domain-specific and less relevant to the task, such as dimensions that describe the position of the robot.

**Imitation learning with a task-relevant embedding of the expert state.** We use the dataset of expert demonstrations provided by Zakka et al. [45] to compare the performance of our approach to that of the XIRL baseline. We follow Zakka et al. [45] and likewise use the simplified imitation

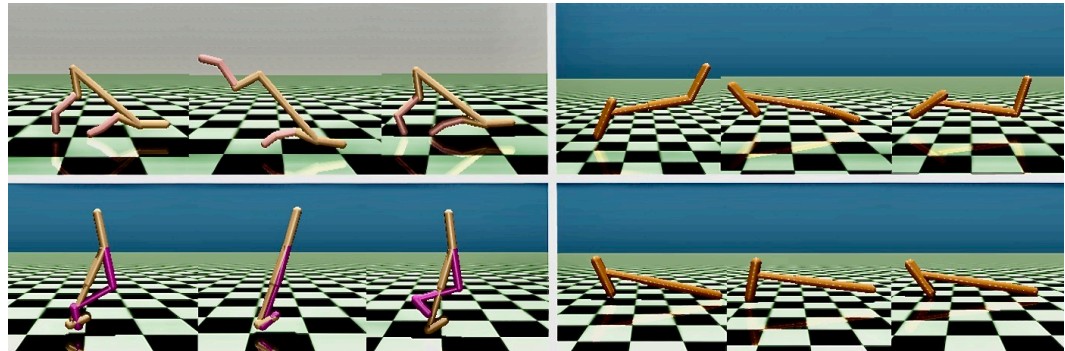

Figure 3: Sample rollouts from the three agents hopper, halfcheetah and walker (section 5.2). We illustrate locomotion strategies learned for different dimensions $d$ of the expert state's embedding space $z$ (see discussion in section 5.3). Right side: For larger $d$, the hopper performs a swimming like movement (top). For smaller $d$ (bottom), the hopper is straight and propels itself forward using only its foot. Left Side: For smaller $d$, the halfcheetah propels itself forward with its front on the ground (top). For larger $d$, the walker performs a mix of a falling and walking motion (bottom).

learning framework where the learner agent simply receives a reward signal that corresponds to the distance between the current environment state and the target environment state, which is pre-computed by averaging over all terminal states contained in the set of expert demonstrations. Note that the main difference between UDIL and XIRL is the task-relevant embedding of the expert state: XIRL relies on the full expert state. We use the XIRL implementation as given by the authors, apply it directly to the state space and do not change any parameters. Figure 2 shows that we consistently outperform XIRL and in both cases achieve a score close to the maximum possible. We find that our method obtains task-relevant embeddings of the state from expert demonstrations alone, which significantly improves performance of cross-domain imitation learning in the XMagical environment.

### 5.2 Cross-domain imitation learning of robot control

We now evaluate UDIL in the complex Mujoco environments [7, 38]. We use the embodiments displayed in Figure 3, hopper, walker and halfcheetah, which are commonly used to evaluate (cross-domain) imitation learning algorithms [20, 16, 12, 30]. We use the fixed-length trajectory implementation [13] of these environments to prevent implicitly rewarding the learner agent for longer trajectories; the significance of this effect is demonstrated in Kostrikov et al. [20]. We first find a minimal task-relevant embedding, investigate the performance, and compare to GWIL. We then conduct ablation studies to evaluate the importance of the individual components of our framework and investigate how the transfer of information from the expert to the learner domains can be controlled by varying the size of the task-relevant expert embedding. We provide videos of the resulting behaviour, as described in in appendix 8.4.

**Finding a task-relevant embedding.** Analogously to the previous section 5.1, we first generate sets of expert and pseudo-random transitions, and compute the mutual information between individual state dimensions and the transition labels. We find that across all three agents, the $x$ position of the torso has highest task-relevance, followed by the $z$ position (height). This intuitively makes sense, as the expert agents receive relatively large rewards during training for moving in the positive $x$ direction, followed by a smaller reward for being in a *healthy* (upright) position [7]. Note here that these findings are derived only from the expert demonstrations, without any knowledge of the rewards. Hereafter, the dimensions which describe the angular positions of the main joints with respect to the torso have highest mutual information; lowest mutual information is found for state dimensions that describe velocities of sub-components. We identify the task-relevant embedding with the elbow method as the positions that describe the torso, and later conduct ablation studies with larger embeddings.

**Jointly learning the learner's policy and mapping function.** We parameterize the learner encoder such that it learns an affine transformation of the input and define its loss as the negative of the

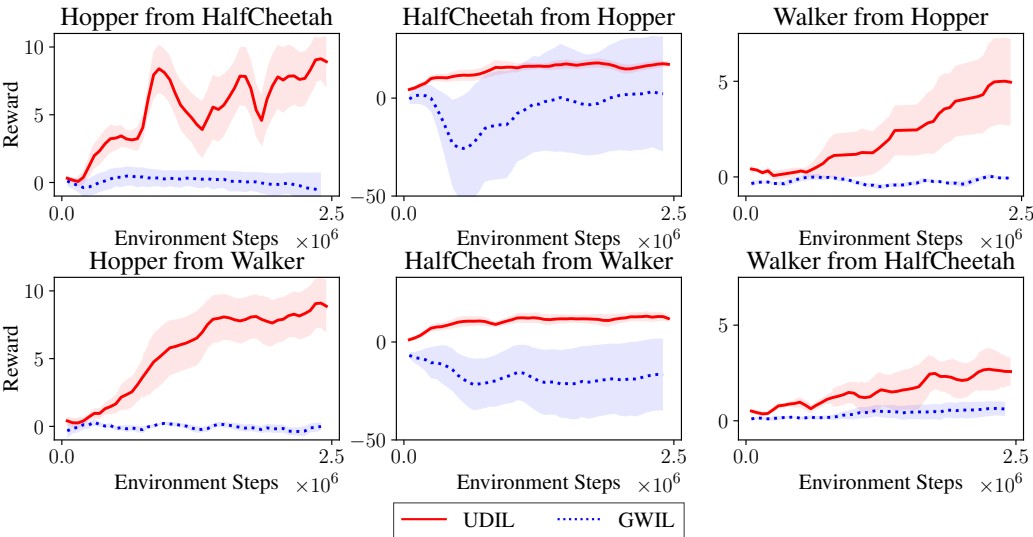

Figure 4: Reward curves for cross-domain imitation learning for different combinations of learner and expert agents. The mean performance is shown as a solid line, and the standard deviation as a shaded area.

discriminator's loss, i.e., the learner encoder is trained to fool the discriminator. The policy of the learner is parameterized by a neural network, which, in contrast to the learner encoder, cannot be trained by backpropagating the discriminator loss as a sampling step is required to obtain the state transitions form the learner policy. We follow Ho and Ermon [16] and train the learner policy with RL, with the learner agent receiving higher rewards for taking actions that result in transformed state transitions $g(s_L), g(s'_L)$ which are more likely to fool the discriminator $D$, i.e., which are more likely to be from the expert's task-relevant state-transition distribution $\rho_E(z_E, z'_E)$. We use DAC [20], to jointly train $g$, $\pi_L$ and $D$, as depicted in Figure 1, and do not alter any hyperparameters given in the original implementation to ensure comparability. We define the reward of the learner agent as the distance covered in the target direction, as this is the only reward component that is common among all three agents, and compare performance to GWIL [10].

**Results.** Figure 4 shows that the learner agents robustly learn meaningful policies for six random initializations across different combinations of expert and learner. We find that the hopper and walker cover about 50% of the distance as compared to when they are trained with their ground truth rewards, with the halfcheetah achieving about 13% of the expert distance.

We qualitatively inspected the behaviours learned by the agents and found novel locomotion strategies that are distinct from those of the expert. We illustrate these strategies in Figure 3. We hypothesize that these new behaviours were enabled by the task-relevant embedding of the expert state and further investigate in section 5.3 how the embedding size can be chosen to transfer more information from the expert to the learner. It can be seen in Figure 4 that our framework consistently outperforms the GWIL baseline; although we tried different hyperparameter configurations, we found the results of GWIL to be highly stochastic, which is due to the properties of the Gromov–Wasserstein distance [25] used, as indicated by the authors of GWIL [10, Remark 1].

### 5.3 Ablation Studies

We present our ablation studies that clarify the importance and influence of the different components of the framework, focusing on the hopper and halfcheetah agents.

**Varying the dimension of the task-relevant embedding.** We investigate the relevance of the task-relevant state embedding's dimension $d$ and hypothesize that for larger embeddings, more information is transferred from the expert to the learner domain. We evaluate the performance as well as the resulting agent behaviours for $d \in (3, 6, all)$, where $all$ refers to no reduction, i.e. $f$ is an identity

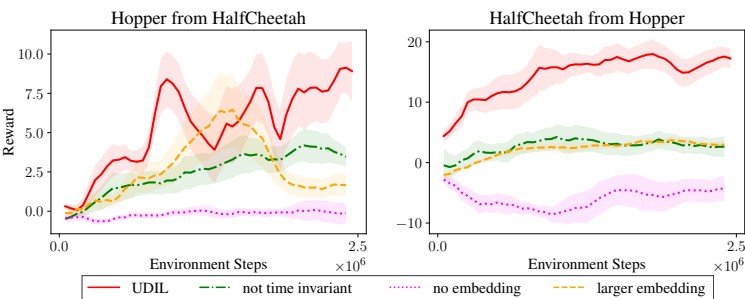

Figure 5: Achieved reward (travelled distance) by both hopper and halfcheetah, when trained on demonstrations of the other with different ablations of our framework. See section 5.3 for details.

mapping, in which case the learner encoder $g$ has to map the full learner state space to the full expert state space. We can observe in Figure 5 that the mean performance and robustness generally decrease when increasing the embedding size. We investigate different locomotion strategies adopted by the learner agent, dependent on the embedding size $d$, and illustrate these in Figure 3. We found that for $d = 3$, both hopper and halfcheetah would lie down on the floor and propel themselves forward. For larger embeddings $d \in \{6, all\}$, both would adopt strategies more similar to the demonstrations by lifting their torso off the ground for longer. The hopper would hop for a few moments and then perform a swimming-like movement, the halfcheetah would exhibit an animal-like quadruped gait.

We conclude that changing the size of the expert's state embedding allows us to modulate the transfer of information between the expert and the learner domains. In one extreme, one might want the learner to solve a task with a minimal task-relevant embedding, to allow the learner to develop strategies distinct from the expert, which could for example allow it to outperform the expert. In the other extreme, one might want the learner to replicate the strategies of the expert as closely as possible, which could be useful if the learner fails to solve the task with less information. Choosing the size of the task-relevant embedding then trades off between these two options.

**Omitting the time invariance constraint.** We omit the time-invariance constraint by reducing the discriminator input from $s, s'$ to just the current state $s$. While this setting yields successful results in same-domain imitation learning [27], we found the time-invariance constraint to be essential for adversarial cross-domain imitation learning (see Figure 5).

**Learning from a single trajectory.** We investigated the performance of our approach when only a single expert trajectory is given, which represents the most direct comparison to GWIL, as GWIL can only utilize a single expert trajectory due to its computational complexity. We find that UDIL likewise outperforms GWIL by a large margin if only one demonstration is given, and show more results in appendix 8.3.3.

# 6 Conclusion

We introduce a novel framework for cross-domain imitation learning, which allows a learner agent to jointly learn to imitate an expert and learn a mapping between both state spaces, when they are dissimilar. This is made possible by defining a mutual information criterion to find a task-relevant embedding of the expert's state, which further allows to control the transfer of information between the expert and learner domains. Our method shows robust performance across different random instantiations and domains, improving significantly upon previous work. However, as cross-domain imitation learning is generally an under-defined problem, the risk of learning incorrect policies remains. The mutual information objective used to find the task-relevant embedding might yield degenerate solutions in special cases, such as when the expert's policy induces a uniform distribution over state transitions, or when the environment is only partially observable. Also, finding the ideal size of the task-relevant embedding might be challenging in more complex domains. Similarly, the application of our algorithm to high-dimensional observation spaces requires further contributions and may constitute an interesting direction for future work.

# 7 Acknowledgements

We thank Dylan Campbell and Jakob Foerster for their helpful feedback. We are also grateful to the anonymous reviewers for their valuable suggestions. This work was supported by the Royal Academy of Engineering (RF\201819\18\163).

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
