# 8 Appendix

## 8.1 Methods

### 8.1.1 IRL Simplification

We first consider the state-only imitation learning objective given in Torabi et al. [39, Equation 7]:

$$\text{IRL}_\psi(\pi_E) = \arg\max_c \left( \min_{\pi_L} \mathbb{E}_{\pi_L} \left[ c(s, s') \right] - \mathbb{E}_{\pi_E} \left[ c(s, s') \right] - \psi(c) \right)$$

We note that the expected cost of a policy can be written as:

$$\mathbb{E}_\pi \left[ c(s, s') \right] = \sum_{s,s'} \rho_\pi(s, s') c(s, s')$$

We assume that the environment state $s$ is composed of $n$ dimensions, i.e. $s = [d_1, d_2, ..., d_n]$. We further assume that the cost function of the expert agent $c_E$ is sparse in the environment dimensions. To simplify notation, we assume that $c_E$ is only a function of the first $m$ dimensions, i.e.

$$c(d_1, d_1', .., d_n, d_n') = c(d_1, d_1'.., d_m, d_m'),$$

where we overload $c$ to take inputs of both dimensionalities. Note that the same reasoning applies to different sparsity patterns without loss of generality. We denote the expert encoder as $f : \mathcal{S}_E \to \mathcal{Z}_E$, mapping the expert state $s_E$ of dimension $n$ to the expert state embedding $z_E$ of dimension $m$. We define $f$ as the operation that truncates the first $m$ dimensions, i.e. it includes all dimensions for which $c_E$ is non-zero. Hence $z = [d_1, .., d_m]$. We can now redefine $c_E$ as a function of $z$. We can then express the expected cost as:

$$\mathbb{E}_\pi[c(s, s')] = \sum_{d_1, d_1', ..., d_m, d_m'} \rho_\pi(d_1, d_1', .., d_m, d_m') \cdot c(d_1, d_1', .., d_m, d_m') \cdot$$

$$\cdot \left( \sum_{d_{m+1}, d_{m+1}', ..., d_n, d_n'} \rho_\pi(d_{m+1}, d_{m+1}', .., d_n, d_n') \right)$$

$$= \sum_{z, z'} \rho_\pi(z, z') \cdot c(z, z').$$

This allows to rewrite the adversarial imitation learning problem as:

$$\text{IRL}(\pi_E) = \arg\max_c \left( \min_{\pi_L} \sum_{z,z'} \rho_{\pi_L}^z(z, z') c(z, z') - \sum_{z,z'} \rho_{\pi_E}^z(z, z') c(z, z') - \psi(c) \right) \quad (8)$$

By exchanging the expert cost function $c_E$ for the expert reward function $r_E$ and flipping the optimization objectives we arrive at equation 4 (which further omits the cost regularizer $\psi$ for reasons of simplicity).

### 8.1.2 Time Invariance Constraint

We consider a 2-dimensional example problem to demonstrate the trivial solutions that can arise when a time-invariance constraint is not imposed on the learner encoder $g$. The expert's embedded state transitions $(z_E^t, z_E^{t+1})$ consist of two numbers drawn from a uniform distribution, obeying $z_E^{t+1} < z_E^t$ (e.g. by rejection sampling).

$$S_E = \left\{ (z_E^t, z_E^{t+1}) : z_E^{t+1} < z_E^t, (z_E^t, z_E^{t+1}) \in [0, 1]^2 \right\} \quad (9)$$

The learner's state transitions $(s_L^t, s_L^{t+1})$ also consist of two numbers drawn from a random distribution, but in contrast $s_L^{t+1} > s_L^t$, i.e. their ordering is reversed.

$$S_L = \left\{ (z_L^t, z_L^{t+1}) : z_L^{t+1} > z_L^t, (z_L^t, z_L^{t+1}) \in [0, 1]^2 \right\} \quad (10)$$

These represent two minimal, but different, distributions to be mapped. We now consider two alternative mapping function domains, one which enforces time-invariance and one which does not. Both are affine functions. The most general, without time-invariance, is

$$g^{\text{affine}}(s_L^t, s_L^{t+1}) = (a \cdot s^{tL} + b, c \cdot s_L^{t+1} + d),$$

parameterized by $a, b, c$ and $d$. A time-invariant specialization of it would be:

$$g^{\text{invariant}}(s_L^t, s_L^{t+1}) = (g'(s_L^t), g'(s_L^{t+1})), \quad g'(s) = a \cdot s + b,$$

which essentially applies the same function $g'$ at both time steps $t$ and $t + 1$.

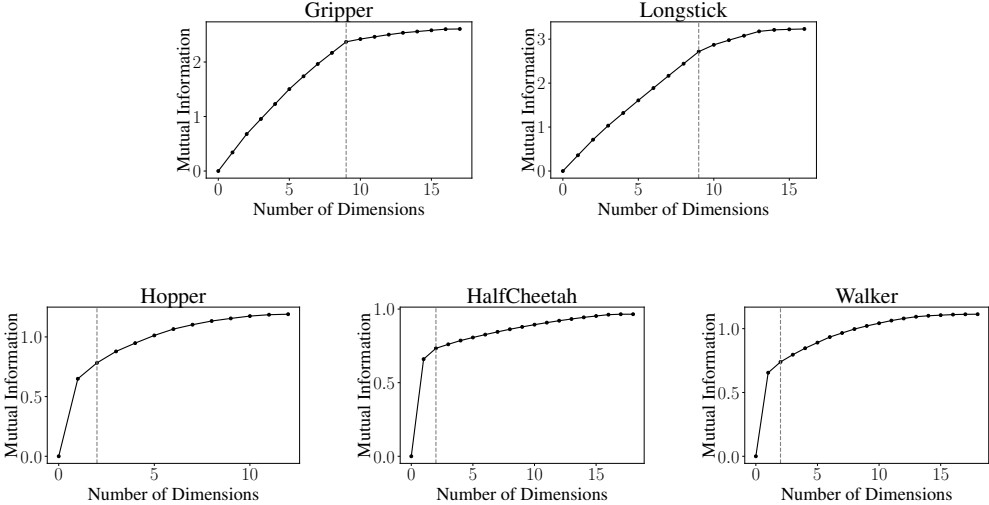

Figure 7: Estimated cumulative mutual information between state transitions $(z, z')$ and labels $(random, expert)$ for increasing size of the expert embedding $z$. The dashed grey line indicates the elbow.

We now analyze the possible solutions that can map $S_E$ and $S_L$ under both models. With $g^{\text{affine}}$, we can simply set $a = c = 0$ (i.e. ignore the input entirely) and $b > d$, to obey the constraint in the learner (eq. 10). This is clearly a trivial solution, since it satisfies the constraint of the output space but ignores the input space entirely (i.e. the output distribution is degenerate).

On the other hand, with $g^{\text{invariant}}$ we cannot set the bias term $b$ independently for different time steps. As a result, the previous trivial solution is not expressible in this model. Instead, we must set $a < 0$ (i.e. negate the input) to map it to the output space while obeying eq. 10.

While this analysis uses a simple model, recall that in practice $g$ is parameterized by a deep network, which are a superset of the set of conforming affine functions. As such, the same trivial solutions must also occur in higher-dimensional settings when time invariance is not enforced.

## 8.2 Experiments

## 8.3 Finding the expert embedding

To find the expert embedding function $f$, we first generate pseudo-random transitions from the set of expert demonstrations, compute the mutual information between the individual state dimensions and the label of a transition (either random or expert) and finally use the elbow method to determine the task-relevant dimensions, which yield the embedding of the expert state.

**Generating sets of random and expert transitions.** We first generate two sets of transitions, one set of expert transitions $\mathcal{T}_E$ and one set of pseudo-random transitions $\mathcal{T}_{rand}$. $\mathcal{T}_E$ is assembled from the transitions contained in the set of expert observations $\mathcal{D}_E$ with a frameskip of 15. We introduce this frameskip to make transitions more distinct, as it ensures that the difference between the two states contained in a transition is substantial. We then generate a set of pseudo-random transitions of the same size as $\mathcal{T}_E$ by randomly sampling two states from $\mathcal{D}_E$ and adding these as a new transition to the set of pseudo-random transitions $\mathcal{T}_{rand}$, until it contains the same number of transitions as $\mathcal{T}_E$.

**Computing mutual information for individual dimensions.** We first compute the estimated mutual information between individual state dimensions and transition labels (random or expert) for which we first define random variables as described in section 4.3 and use the method of Ross [31] to compute the mutual information for each state dimension $n$, arriving at a vector of size $n$ that describes the mutual information between a transition in each state dimension and the label.

**Finding the task-relevant dimensions with the elbow method.** We now compute the cumulative mutual information for all $k \in \{0, .., n\}$ by summing up the mutual information of the $k$ dimensions with largest information. This is plotted in Figure 7. We use the implementation of Satopaa et al. [33] to find

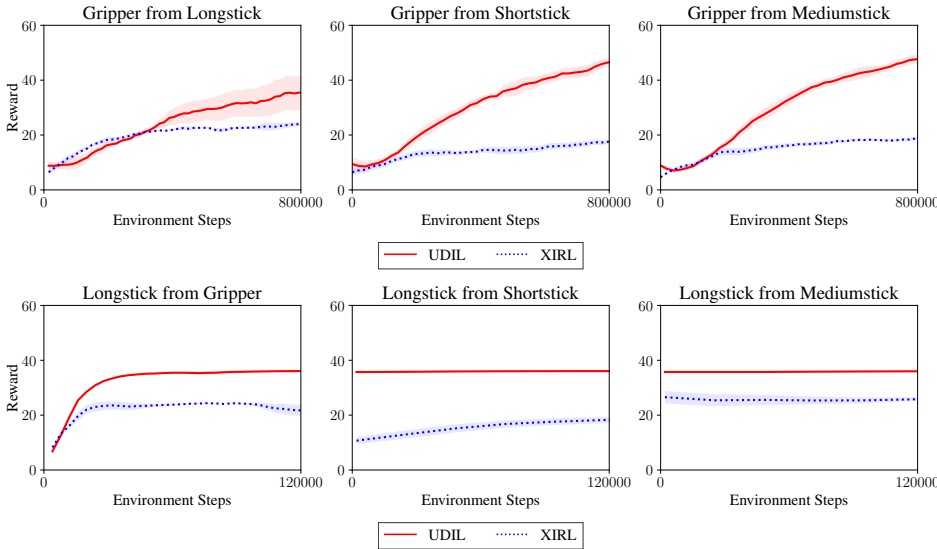

Figure 8: We evaluate the reward achieved by both learner agents when trained on demonstrations of either one of the remaining three embodiments, using either our algorithm UDIL, or the XIRL [45] baseline.

the elbow in the curve, a method commonly used to identify the number of clusters for dimension reduction [19]. The found elbows are likewise displayed in Figures 7. We then estimate the objective stated in eq. 6, i.e. $\arg\max_f I((Z, Z'); Y)$, by defining $f$ such that is reduces the expert state $s_E$ to those dimensions top the left of the elbow, including the elbow itself.

**Background on elbows found.** For XIRL (see sec. 5.1), the task-relevant embedding dimensions found, i.e. those to the left of the elbow, are those 9 dimensions that describe the task-relevant objects. That is, these dimensions describe the three $x$ positions of the blocks seen in Figure 2 (left), the three $y$ positions and the distances between the objects and the target zone. In the Gym environments hopper, walker and halfcheetah (see sec. 5.2), the found task-relevant dimensions describe properties of the torso. That is, for the hopper, they describe the $x$ and the $z$ position of the torso, for the halfcheetah they describe the $x$ coordinate of the torso and the $x$ coordinate of the front tip, and for the walker they describe the $x$ coordinate of the torso and the velocity of the torso in $x$ direction.

### 8.3.1 XIRL Experiments

**Setup.** We use the X-Magical environment [45, 40], as implemented by the authors. [3] We further use the XIRL [45] baseline implementation as implemented by the authors. [4] We use the agents $gripper$ and $longtstick$, as these have the largest difference in embodiment. In contrast to XIRL, we only train on demonstrations of one other agent. We do not use the pixels as observations, but use the environment state vector directly. We increase training time by a factor of two, as we found that convergence was not reached otherwise, and leave all other parameters unchanged. We evaluate UDIL and XIRL for six different random seeds and report mean and standard error in Figure 2.

**Results for additional embodiments.** We further evaluated both UDIL and XIRL on demonstrations of the remaining embodiments of the X-Magical benchmark [41, 45]. Results for the embodiments $Gripper$ and $Longstick$, trained cross-domain from demonstrations from three of the four given embodiments ($Gripper$, $Longstick$, $Shortstick$, $Mediumstick$) are shown in Figure 8. We find that UDIL outperforms XIRL consistently across all tested pairings of embodiments.

**Results for UDIL with adversarial training.** We further evaluated both the simplified version of UDIL (which, analogously to XIRL [45], rewards the agent for minimizing the distance to the pre-computed goal state), and the performance of the original implementation of UDIL (see eq. 7) that uses adversarial training. It can be observed in Figure 9 that the adversarial implementation of UDIL outperforms the XIRL baseline in both cases.

---

[3] https://github.com/kevinzakka/x-magical
[4] https://x-irl.github.io

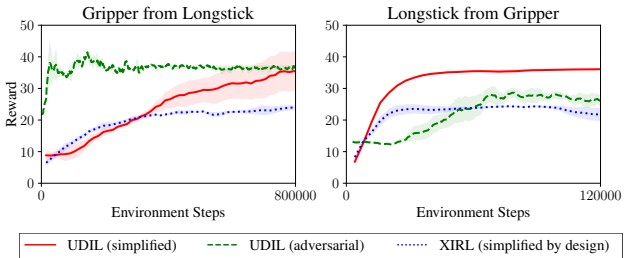

Figure 9: We evaluate the reward achieved by both learner agents when trained on demonstrations of the other, using either the simplified version of UDIL, the unmodified adversarial version of UDIL, or the XIRL [45] baseline, which uses a simplified implementation by design.

Table 1: Hyperparameters used to train learner encoder $g$.

|  | Hopper | HalfCheetah | Walker |
| --- | --- | --- | --- |
| Learning rate encoder ($\alpha$-enc) | 0.001 | 0.001 | 0.0001 |
| Use bias with encoder (enc-use-bias) | False | True | False |
| Train every $n$-enc steps | 0.01 | 0.01 | 0.1 |

However, it performs inconsistently with respect to the simplified version of UDIL (once performing better, once worse).

### 8.3.2 Gym Experiments

**Setup.**  We train the learner policy $\pi_L$, the mapping $g$ between the learner agent's states $s_L$ and the expert agent's task-relevant state embedding $z_E$, and the discriminator $D$ jointly (see blue components in Figure 1). We reimplement the discriminator-actor-critic algorithm [21], resembling the original implementation given by the authors as close as possible, [5]. We keep all parameters unchanged and refer to the original implementation for further details. We further use the StableBaselines3 [6] package to implement the reinforcement learning agents and the Seals package [7] to implement the gym environments with fixed episode length. We do not alter any parameters given in these implementations.

We introduce a minimal set of additional hyperparameters that all regard the learner encoder $g$, which are given in Table 1. We appended the discriminator-actor-critic framework by the expert encoder $g$ (described in the next section), which is trained by backpropagating the negative discriminator loss, i.e. the encoder $g$ is trained to fool the discriminator $D$. We train the learner encoder $g$ every $n$-encoder steps of the discriminator, i.e. the encoder is trained less frequently than the discriminator, and use a learning rate $\alpha$-enc. We train the learner agent with 20 expert trajectories, which were generated by an expert agent trained with the ground truth reward in the respective environment. We run each experiment for six seeds (zero to five) to ensure robustness to different random instantiations and report the mean and standard error in Figure 4.

**Learner Encoder.**  We parameterise the learner encoder $g$ such that it learns an affine transformation, i.e. it applies an affine transformation to the learner state $s_L$. To stabilize learning, we apply a *sigmoid* that scales the transformation weights (and the bias), such that they do not exceed a maximum magnitude of five. The learner encoder $g$ is implemented as a single layer neural network that outputs a weight for each input dimension, which may be appended by a bias (indicated by *enc-use-bias*).

**GWIL Baseline.**  We run the GWIL baseline [10] using the authors implementation. [8] We evaluated different combinations for the hyperparameters *gw-entropic* and *gw-normalize* and found that the author's original implemtation worked best. We evaluated the baseline likewise for the random seeds zero to five and report mean and standard error in Figure 4. We found results to be highly stochastic, to the extent that not a single positive result was achieved in some, as also described by the authors [10, Remark 1].

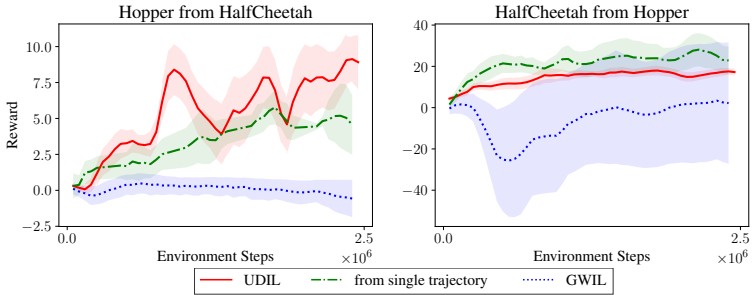

Figure 10: Achieved reward (travelled distance) by both hopper and halfcheetah, when trained on only a single demonstrations of the other. See section 5.3 for details.

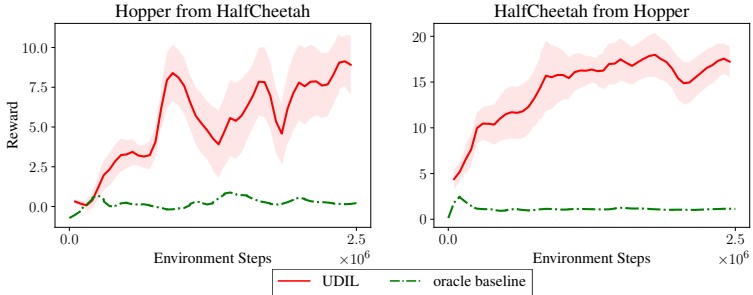

Figure 11: Achieved reward (travelled distance) by both hopper and halfcheetah, when trained with an oracle approach that omits the learner encoder $g$. See section 8.3.3 for details.

### 8.3.3 Ablation Studies

**Imitation from a single demonstration.** We evaluated the performance of UDIL when only a single expert demonstration (single trajectory) is given. This constitutes the closest comparison to GWIL, as it does not scale to more than one trajectory due to its computational complexity. We can observe in Figure 11 that UDIL also outperforms GWIL if only a single trajectory is given. We further find that the performance of the halfcheetah, when imitating the hopper, is higher for one trajectory (as compared to the usual 20 trajectories). We further investigated this and found it to be an outlier, as this was not the case for any other agent combination.

**Comparison to an oracle baseline.** We further compared the performance of UDIL to that achieved by an oracle baseline, designed as follows. We assume that an oracle is used to choose the state dimensions of the learner agent which match those of the expert included in the task relevant embedding, while the order of the states remains unknown. We then run UDIL directly on the task-relevant embedding, i.e. omitting the learner encoder $g$.

### 8.4 Videos

We provide videos of the resulting behaviours in both XMagical and Gym in the supplementary material.

---

[5] https://github.com/google-research/google-research/tree/master/dac
[6] https://github.com/DLR-RM/stable-baselines3
[7] https://github.com/HumanCompatibleAI/seals
[8] https://github.com/facebookresearch/gwil