# OpenReview forum: "Learn what matters: cross-domain imitation learning with task-relevant embeddings"
_NeurIPS.cc/2022/Conference — NeurIPS 2022 Accept_

### Official Review · Reviewer_YPHf · 2022-07-10

**Rating:** 7
**Confidence:** 4
**Soundness:** 3 good
**Presentation:** 3 good
**Contribution:** 3 good

**Summary:**

The problem setting under consideration is cross-domain imitation learning, where the goal is to enable an imitation learning agent to learn from expert demonstrations from a different environment or agent embodiment. The authors propose a method for cross-domain imitation learning that requires less supervision than prior methods (i.e. without additional proxy tasks or multiple demonstrators/domains), by primarily leveraging a mutual information-based objective to encourage learning a more task-relevant representation of the expert state space. This embedding is then used by the learner for an imitation learning objective. The method further imposes a time-invariance constraint to prevent learning a degenerate embedding space, and the overall method uses an adversarial imitation learning setup to learn solely from observations. The experiments compare against other recent methods in cross-domain imitation learning, across a range of task settings.

**Questions:**

- Have the authors considered / experimented with using multiple expert agents providing demonstrations? One of the stated motivations for this work is wanting to avoid reliance on a large number of different demonstrators, but it would be interesting to see if there is any performance improvement in the multi-demonstrator case (i.e. would the quality of the learned embedding space improve, or is one demonstrator sufficient for capturing the task-relevant features?).
- In the ablations plot (Figure 5) with Hopper from HalfCheetah, what is the intuition behind the noisy reward curve for UDIL? As adversarial training objectives can be unstable to train, I am curious if the authors have seen any other similar instabilities / difficulties with training.
- Do the authors have any hypotheses for why the performance increases then decreases for the larger embedding case for Hopper from HalfCheetah in Figure 5?

**Limitations:**

The paper would be improved with a section more clearly discussing the limitations of the proposed approach -- e.g. if there are difficulties with adversarial training, whether there are cross-domain environments or tasks where the proposed mutual information objective would fail, or how much additional overhead is required to search for the best embedding dimension size for the task you care about.

**Strengths And Weaknesses:**

Strengths:
- Observational cross-domain imitation learning has grown increasingly relevant as a problem setting, for which the authors propose a novel self-supervised method (to the best of my knowledge). While the individual components -- mutual information based representation learning, adversarial imitation learning -- are not novel, the construction of the method and problem domain seems novel. This problem setting and proposed method has high relevance to the research community, as methods developed in this area opens the door for unsupervised learning of behaviours from widely available online demonstrations (eg. Youtube videos).
- The method is well explained and straightforward. The authors carry out a range of evaluations and ablations to show the capabilities of their proposed method, comparing against recent work in this domain, and show that their proposed method performs well.
- The paper is very well written, structured nicely, and was a pleasure to read.
- Being able to adjust how much of the task-relevant information to retain using the embedding size is an interesting outcome of the method.

Weaknesses:
- While not explicitly directed at imitation learning across different embodiments, there are some relevant works in unsupervised methods for domain regularization in observational imitation learning, which should be cited in the paper:
  - Stadie, Bradly C., Pieter Abbeel, and Ilya Sutskever. "Third-person imitation learning." (2017).
  - Cetin, Edoardo, and Oya Celiktutan. "Domain-robust visual imitation learning with mutual information constraints." (2021).
- While the performance in Section 5.1 compared to XIRL seems strong, I would like to see the same evaluations carried out on different combinations of agent embodiments rather than just the two that are most different. In the other embodiment cases, does UDIL still outperform XIRL or is the performance more similar as the embodiment gap closes?
- It also seems like the authors do not use the adversarial imitation learning setup in the comparisons against XIRL. It would be interesting to see if the adversarial setup improves or hurts performance in this case.

---

> ### Author Response · Authors · 2022-08-02
> **Response to Reviewer**
>
> >Strengths And Weaknesses:
> > Strengths:
> > - Observational cross-domain imitation learning has grown increasingly relevant as a problem setting, for which the authors propose a novel self-supervised method (to the best of my knowledge). While the individual components -- mutual information based representation learning, adversarial imitation learning -- are not novel, the construction of the method and problem domain seems novel. This problem setting and proposed method has high relevance to the research community, as methods developed in this area opens the door for unsupervised learning of behaviours from widely available online demonstrations (eg. Youtube videos).
> > - The method is well explained and straightforward. The authors carry out a range of evaluations and ablations to show the capabilities of their proposed method, comparing against recent work in this domain, and show that their proposed method performs well.
> > - The paper is very well written, structured nicely, and was a pleasure to read.
> > - Being able to adjust how much of the task-relevant information to retain using the embedding size is an interesting outcome of the method.
>
> > Weaknesses:
>
> > - While not explicitly directed at imitation learning across different embodiments, there are some relevant works in unsupervised methods for domain regularization in observational imitation learning, which should be cited in the paper:
> >     - Stadie, Bradly C., Pieter Abbeel, and Ilya Sutskever. "Third-person imitation learning." (2017).
> >     - Cetin, Edoardo, and Oya Celiktutan. "Domain-robust visual imitation learning with mutual information constraints." (2021).
>
> Thank you for this input. We added these works to the related works section.
>
> > - While the performance in Section 5.1 compared to XIRL seems strong, I would like to see the same evaluations carried out on different combinations of agent embodiments rather than just the two that are most different. In the other embodiment cases, does UDIL still outperform XIRL or is the performance more similar as the embodiment gap closes?
>
> Thank you for this suggestion. We ran these experiments and added a comparison of the performance of UDIL and XIRL also for the remaining two embodiments (mediumstick and shortstick). Please see section 7.3.1 or Figure 8 for the results in the updated paper. We found that UDIL outperforms XIRL consistently over all tested configurations. Interestingly, the performance gap between UDIL and XIRL was even larger for the newly added scenarios.
>
> > - It also seems like the authors do not use the adversarial imitation learning setup in the comparisons against XIRL. It would be interesting to see if the adversarial setup improves or hurts performance in this case.
>
> Thank you for this suggestion. We have added this evaluation in section 7.3.3. (”Results for UDIL with adversarial training.”, see Figure 9). We find that the unchanged implementation of UDIL that uses adversarial training outperforms the XIRL baseline in both scenarios.
>
> > Questions:
>
> > - Have the authors considered / experimented with using multiple expert agents providing demonstrations? One of the stated motivations for this work is wanting to avoid reliance on a large number of different demonstrators, but it would be interesting to see if there is any performance improvement in the multi-demonstrator case (i.e. would the quality of the learned embedding space improve, or is one demonstrator sufficient for capturing the task-relevant features?).
>
> We have not done such an evaluation yet, but do agree that it would be highly interesting and we share the intuition that demonstrations from distinct demonstrators might improve the task-relevant embedding. In this sense, we see this as interesting direction for future work.
>
> > - In the ablations plot (Figure 5) with Hopper from HalfCheetah, what is the intuition behind the noisy reward curve for UDIL? As adversarial training objectives can be unstable to train, I am curious if the authors have seen any other similar instabilities / difficulties with training.
>
> We agree with the intuition that the noisy reward curve should be an artefact of the adversarial training objective. A better choice of hyper-parameters for the adversarial imitation learning algorithm might improve this aspect, but we instead used the hyper-parameter settings of the original authors to ensure comparable results.
>
> > - Do the authors have any hypotheses for why the performance increases then decreases for the larger embedding case for Hopper from HalfCheetah in Figure 5?
>
> We hypothesise that this is due to the different locomotion modes that the Hopper can adopt, i.e. we hypothesise that the hopper changes its locomotion mode to one that is more similar to the expert embedding, which however yields lower returns (travels less far).

---

> > ### Comment · Reviewer_YPHf · 2022-08-08
> > **Response to Rebuttal**
> >
> > Thank you for the thorough response and additional evaluations. I will be keeping the original score (7).

---

### Official Review · Reviewer_LDkQ · 2022-07-11

**Rating:** 6
**Confidence:** 4
**Soundness:** 3 good
**Presentation:** 4 excellent
**Contribution:** 3 good

**Summary:**

The authors propose to address cross-embodiment imitation learning by using GAILfO but with a learned cross-embodiment embedding.  They learn the mappings from (i) expert state space to the embedding space, and (ii) learner state space to the embedding space by respectively (i) using a mutual-information objective with state transition pairs and psuedorandom state transition pairs, and (ii

**Questions:**

Please see my discussion of Weaknesses for questions.


**Limitations:**

The authors mention only that "the risk of learning incorrect policies remains".  I encourage the authors to consider and discuss other limitations that I raised in the Weaknesses discussion.

**Strengths And Weaknesses:**

## Strengths

1. The formulation is pretty clean, satisfying, and nice to follow.

2. The results show that the authors are able to get the method to work pretty well on the challenging problem of cross-embodiment imitation.

## Weaknesses

1. Discussion of relation to existing works
  - Issue: The discussion of prior work is kind of annoyingly pedantic, and doesn't have to be, in attempting to carve out a novelty statement about what other methods can or can't do.  In general, paragraph 4 of the intro (continuing onto page 2), and the "cross-domain imitation learning" discussions in the Related Work, and the brief mention of related work in the Abstract, are all annoying. For example, they say that [40] needs multiple demonstrator agents, but as they show themselves, it is pretty trivial to just use [40] but with a single demonstrator agent.  Also, [9] is another prior work that effectively addresses the same problem statement.  Also, some of the limitations of it, such as the footnote on page 2, also apply to the presented work.  To be honest, I'm not sure the authors are able to say any type of clean statement about a type of scenario that their algorithm can address that others can't, and I think they know that.  But that's okay!  That doesn't mean their work isn't interesting or isn't useful.
  - Suggested solution: I think the authors would do well to focus on: rather than trying to come up with statement of what other methods can't address (which has some issues as noted above), they might do well to just instead focus on how their proposed formulation is *different*, how it is interesting, and how at least in the shown experiments it may also work better.  Additionally, they could be more positive towards, and appreciative of, the prior works in this challenging subfield, which have helped carve a path that they can follow and attempt to improve upon.  Rather than their current final sentence on lines 81-84, they could basically just say "We propose a different formulation using X, Y, and Z, and further show that compared to [40] and [9], empirical results suggest our method is more capable than these prior works on the tested settings."  Or something like that.

2. Is "cross-domain" better terminology to be using here than "cross-embodiment"?  All of the experiments are focused on cross-embodiment imitation learning, rather than any other notion of difference between expert and learner.  I don't think it matters too much either way, but as the authors themselves say, the cross-embodiment case is probably in general the most challenging, and hence why they focus on it.

3. Probably a different title should be necessary?  Per my discussion of point 1 above, there are actually prior works which address the cross-embodiment imitation learning setting, and similarly do so with the same amount of supervision as the present work (only expert demonstrations needed from a different embodiment).  Accordingly, I don't think the title is very truthful or accurate.  It seems to suggest that this paper introduces the idea of doing cross-domain imitation learning, which as discussed in point 1, isn't the case.  Further, it's not really unsupervised... there isn't really such a thing as unsupervised imitation learning.  Specifically, the expert is providing supervision.  A more accurate and useful title for the community might be something like "Generative Adversarial Cross-Embodiment Imitation with Task-Relevant Embeddings" or something like that.  That title actually calls out how the paper addresses a known problem, but with a different method.

4. Probably should take one of the contributions off the list?  Specifically the first stated contribution, "We devise a framework to learn the mapping between the learner and expert domains in an unsupervised fashion, i.e., without additional proxy task demonstrations." also applies to for example [40] and [9], so I don't think it can be claimed as a contribution of the present work.  However, I think the other 2 contributions in the list are strong and sufficient.

5. Some cases where the proposed algorithm definitely fails. For example consider a case where of simple environment where an agent visits every state, in a particular order, exciting every possible state-next-state $(s, s')$ transition an equal number of times.  In this case, the task-relevant embedding would fail to do anything useful, since although the long-horizon order may be considered what matters, instead the algorithm only uses state-next-state pairs to find the task-relevant embedding.  Although this is a simple case, it may in general point to problems for using the proposed algorithm on long-horizon, multi-step tasks.

6. Some cases where the proposed algorithm I think would fail.  On this point, I may be wrong, but I just don't see how specifically the learning of $g$ (the mapping from learner state to shared embedding space) is required to learn anything useful.  I just don't see what's stopping $g$ from deciding that it wants to completely scramble the ordering of all states visited.  While Section 4.4 discusses avoiding a degenerate mapping for $f$, it doesn't discuss it for $g$.

7. Some missing key related work.  Another very related work is "Reinforcement Learning with Videos" (https://arxiv.org/pdf/2011.06507.pdf).  Although I think it still significantly different from the proposed work, for example not adopting the GAIL style formulation, it does also address cross-embodiment imitation and also learns a shared embedding space.  Note too that the v2 November 2021 version actually corrected a prior mistake which was that, contrary to the published conference version, the method does not actually need paired data.  Accordingly it has, like the other works mentioned too, basically the same data requirements as the presented work.

8. Issue with time-invariant mapping not discussed.  This related to my point #6.  If f is time-invariant, it can't know about histories of states mattering in any particular way, which makes long-horizon demonstrations with potentially repetitive tasks not addressable.  Also, they don't discuss how the actually make it time invariant.  It already looks time invariant in Section 4.3 -- seems like they don't need to do anything else to make it so.

9. Note that the comparison between the present work and Kim et al [17] could be improved.  Basically the present work discards [17] because it needs some proxy tasks.  But could the authors describe why specifically they need proxy tasks?  This is an especially important comparison because Kim et al generally falls in the category of doing cross-embodiment generative advsersarial imitation.  So the details really matter here with the comparison, I think.


## Minor

1. Line 23 says that "classic" imitation learning algorithms do state occupancy matching between learner and expert... this though is the formulation of GAIL?  I'm not sure I would call GAIL a "classic" IL method yet?

2. Eq. 5 would probably be even more clear if rather than $z_E$ it was called out that this is $f(s_E)$.  Then it's clear that both expert and learner state spaces are being mapped to some shared embedding space.

3. Error: line 186, first symbol, should be $g$.

4. Lines 104-106 -- they make it seem like an extra-special thing to not know the actions of the expert, but in the cross-embodiment case, the learner has different actions anyway, so it's kind of already implied that the actions won't be that useuf.

5. Line 109 -- Torabi et al, GAILfO, is a good paper, but as those authors would probably admit themselves, as is highlighted in the name of their algorithm, it is a pretty minor modification to GAIL.  Accordingly, it seems justified to give a citation to GAIL as well as GAILfO on line 109.

6. Error: line 154 should say $g: S_L \rightarrow Z_E$.

7. May consider just using $Z$ for notation, rather than $Z_E$, since the embedding space is used by both learner and expert, rather than just expert.

8. Line 230-233 is confusing -- who is "we" here?  Are you describing how you modify [40], or what the proposed algorithm uses?

9. Line 241 -- I'm not sure anybody would call hopper, walker, and half-cheetah "high-dimensional"?  I think walker has a 3D action space, and the others are 6D?  6D isn't small... it's moderately hard... but I don't think "high-dimensional".

---

> ### Author Response · Authors · 2022-08-02
> **Response to Reviewer (part 1)**
>
> >1. Discussion of relation to existing works
> > - Issue: The discussion of prior work is kind of annoyingly pedantic, and doesn't have to be, in attempting to carve out a novelty statement about what other methods can or can't do. In general, paragraph 4 of the intro (continuing onto page 2), and the "cross-domain imitation learning" discussions in the Related Work, and the brief mention of related work in the Abstract, are all annoying. For example, they say that [40] needs multiple demonstrator agents, but as they show themselves, it is pretty trivial to just use [40] but with a single demonstrator agent. Also, [9] is another prior work that effectively addresses the same problem statement. Also, some of the limitations of it, such as the footnote on page 2, also apply to the presented work. To be honest, I'm not sure the authors are able to say any type of clean statement about a type of scenario that their algorithm can address that others can't, and I think they know that.  But that's okay! That doesn't mean their work isn't interesting or isn't useful.
> > - Suggested solution: I think the authors would do well to focus on: rather than trying to come up with statement of what other methods can't address (which has some issues as noted above), they might do well to just instead focus on how their proposed formulation is *different*, how it is interesting, and how at least in the shown experiments it may also work better. Additionally, they could be more positive towards, and appreciative of, the prior works in this challenging subfield, which have helped carve a path that they can follow and attempt to improve upon. Rather than their current final sentence on lines 81-84, they could basically just say "We propose a different formulation using X, Y, and Z, and further show that compared to [40] and [9], empirical results suggest our method is more capable than these prior works on the tested settings." Or something like that.
>
> Thank you for this feedback. It was by no means our intention to understate the importance of prior work in the field. We solely intended to point out the differences between ours and previous work with as much detail as possible, while highlighting potential advantages of our approach. We implemented the given suggestions, trying to balance the required detail needed for readers not familiar with the field.
> More specifically, we stated more clearly that the problem setting considered by GWIL ([9]) is equivalent to ours and that the approach of XIRL ([40]) can be directly adapted to be applicable to our problem setting.
>
> > 2. Is "cross-domain" better terminology to be using here than "cross-embodiment"? All of the experiments are focused on cross-embodiment imitation learning, rather than any other notion of difference between expert and learner. I don't think it matters too much either way, but as the authors themselves say, the cross-embodiment case is probably in general the most challenging, and hence why they focus on it.
>
> We agree that our experiments focus on cross-embodiment imitation, which we now stated more clearly in the introduction. As the approach is not limited to cross-embodiment applications but is stated generally enough to address other domain mismatches, we chose the term cross-domain instead of cross-embodiment (also following the nomenclature in recent literature).
>
> > 3. Probably a different title should be necessary? Per my discussion of point 1 above, there are actually prior works which address the cross-embodiment imitation learning setting, and similarly do so with the same amount of supervision as the present work (only expert demonstrations needed from a different embodiment). Accordingly, I don't think the title is very truthful or accurate. It seems to suggest that this paper introduces the idea of doing cross-domain imitation learning, which as discussed in point 1, isn't the case. Further, it's not really unsupervised... there isn't really such a thing as unsupervised imitation learning. Specifically, the expert is providing supervision. A more accurate and useful title for the community might be something like "Generative Adversarial Cross-Embodiment Imitation with Task-Relevant Embeddings" or something like that. That title actually calls out how the paper addresses a known problem, but with a different method.
>
> Thank you for this remark, we agree that the term “unsupervised” can be overloaded. We use it in the strictest possible sense afforded by imitation learning: even though in imitation learning demonstrations of expert behaviour are always needed, our approach does not require additional demonstrations from proxy tasks, or a reward signal. We now stated this more clearly in section 4. We hence agree that the previous title could cause confusion and would change it to “Learning what matters: cross-domain imitation learning with task-relevant embeddings”.

---

> ### Author Response · Authors · 2022-08-02
> **Response to Reviewer (part 2)**
>
> > 4. Probably should take one of the contributions off the list? Specifically the first stated contribution, "We devise a framework to learn the mapping between the learner and expert domains in an unsupervised fashion, i.e., without additional proxy task demonstrations." also applies to for example [40] and [9], so I don't think it can be claimed as a contribution of the present work. However, I think the other 2 contributions in the list are strong and sufficient.
>
> Thank your for this valid critique. We have rephrased the contributions accordingly.
>
> > 5. Some cases where the proposed algorithm definitely fails. For example consider a case where of simple environment where an agent visits every state, in a particular order, exciting every possible state-next-state (s,s′) transition an equal number of times. In this case, the task-relevant embedding would fail to do anything useful, since although the long-horizon order may be considered what matters, instead the algorithm only uses state-next-state pairs to find the task-relevant embedding. Although this is a simple case, it may in general point to problems for using the proposed algorithm on long-horizon, multi-step tasks.
>
> Thank you for pointing this out. We agree that no meaningful embedding can be found if the expert’s policy induces a uniform state-next-state distribution. Even though this specific scenario seems rather unlikely in practice, we included it in our discussion of limitations in the conclusion, as it has theoretical relevance. We also acknowledge that the mutual information objective might yield degenerate solutions in the case of long-horizon tasks, especially if the the environment state is only partially observed or the observations are noisy, and address this in the conclusion.
>
> > 6. Some cases where the proposed algorithm I think would fail. On this point, I may be wrong, but I just don't see how specifically the learning of g(the mapping from learner state to shared embedding space) is required to learn anything useful. I just don't see what's stopping g from deciding that it wants to completely scramble the ordering of all states visited. While Section 4.4 discusses avoiding a degenerate mapping for f, it doesn't discuss it for g.
>
> We agree that it was unclear why no time-invariance constraint was used to learn the expert encoder f. In fact, our problem formulation in section 4.3 results in a time-invariant encoder f, which we now state explicitly.
>
> > 7. Some missing key related work. Another very related work is "Reinforcement Learning with Videos" (https://arxiv.org/pdf/2011.06507.pdf). Although I think it still significantly different from the proposed work, for example not adopting the GAIL style formulation, it does also address cross-embodiment imitation and also learns a shared embedding space. Note too that the v2 November 2021 version actually corrected a prior mistake which was that, contrary to the published conference version, the method does not actually need paired data. Accordingly it has, like the other works mentioned too, basically the same data requirements as the presented work.
>
> Thank you for this suggestion, we now added this reference to our literature overview. We initially omitted this work in the related work section because it assumes that both the correct reward signal and demonstrations of the task are given (effectively combining reinforcement learning and imitation learning). This setting is different to ours, as we do not assume access to the reward signal. This is also highlighted by the authors of the given work, which state that the demonstrations primarily act to speed up exploration, which is a problem formulation different from ours. However, we have now cited it.
>
> > 8. Issue with time-invariant mapping not discussed. This related to my point #6. If f is time-invariant, it can't know about histories of states mattering in any particular way, which makes long-horizon demonstrations with potentially repetitive tasks not addressable. Also, they don't discuss how the actually make it time invariant. It already looks time invariant in Section 4.3 -- seems like they don't need to do anything else to make it so.
>
> Please see the answer to question 5. There was also an error in our notation in section 4.3, which we now corrected.

---

> ### Author Response · Authors · 2022-08-02
> **Response to reviewer (part 3)**
>
> > 9. Note that the comparison between the present work and Kim et al [17] could be improved. Basically the present work discards [17] because it needs some proxy tasks. But could the authors describe why specifically they need proxy tasks? This is an especially important comparison because Kim et al generally falls in the category of doing cross-embodiment generative advsersarial imitation. So the details really matter here with the comparison, I think.
>
> The work of Kim et al. uses the proxy task demonstrations to first learn a mapping between the two domains, which is then used to find the policy in the subsequent step. Our approach combines these two steps, thereby omitting the need for demonstrations of proxy tasks in both domains.
>
> > Minor
>
> > 4. Lines 104-106 -- they make it seem like an extra-special thing to not know the actions of the expert, but in the cross-embodiment case, the learner has different actions anyway, so it's kind of already implied that the actions won't be that useuf.
>
> Thank you for this remark. We pointed this out as related work (e.g. GWIL [9]) also uses the actions of the expert agent, which is different from our approach.
>
> > 7. May consider just using Z for notation, rather than ZE, since the embedding space is used by both learner and expert, rather than just expert.
>
> Thank you for this suggestion, we updated our notation accordingly.
>
> > 8. Line 230-233 is confusing -- who is "we" here? Are you describing how you modify [40], or what the proposed algorithm uses?
>
> We follow XIRL here, thank you for pointing this out, we made it more clear now.
>
> > Limitations
>
> > The authors mention only that "the risk of learning incorrect policies remains". I encourage the authors to consider and discuss other limitations that I raised in the Weaknesses discussion.
>
> We now revised our conclusion accordingly.

---

> > ### Comment · Reviewer_LDkQ · 2022-08-08
> > **Reviewer response to rebuttal**
> >
> > Thanks, very thorough response. I don't know if you just made all the changes just because I requested them, or if you actually think it makes the paper better, but regardless, personally I do think they all make the paper much better.  Upgrading my score as follows
> >
> > (note for posteriority):
> > previous overall rating: 5
> > updated overall rating: 6
> >
> > previous presentation: 3, good
> > updated presentation: 4, excellent

---

### Official Review · Reviewer_KU89 · 2022-07-11

**Rating:** 6
**Confidence:** 3
**Soundness:** 3 good
**Presentation:** 4 excellent
**Contribution:** 3 good

**Summary:**

The paper presents UDIL, unsupervised cross-domain imitation learning, a method for learning a policy in one environment using expert demonstrations from another environment. The focus is on learning a task-relevant embedding, which identifies which parts of the state should be used for mapping between learner and expert data. Experiments are presented in two domains: the XMagical benchmark (in which the learner and expert are circular and a long stick or vice versa) and MuJoCo (in which the agents are hopper, halfcheetah, or walker). UDIL outperforms XIRL in the XMagical domain and outperforms GWIL in the MuJoCo domain.

**Questions:**

I wonder if there are additional baselines that could be used. For example, what happens if vanilla imitation learning is done on the task-relevant embedding? I expect this would work poorly since the policy does need some environment-specific information. Another baseline could be some form of oracle in which the task-relevant dimensions are hand-picked or make use of the reward in some way.

Minor (no need to respond):
* Typo on line 293 (missing period after domain).
* Typo on line 318 (3 p's in appendix).

**Limitations:**

I'd like to see a discussion of limitations, perhaps addressing how the method might extend to more complex observations/environments.

**Strengths And Weaknesses:**

Strengths
* The paper is well-written and well-motivated. The method and design decisions are explained clearly.
* The analysis of the size of the task-relevant embedding (dimension d) is interesting and original.

Weaknesses
* The experiments are limited to environments with small states, where there is a clear distinction between task-relevant dimensions and dimensions which can be discarded. The claims would be strengthened if experiments were extended to environments with visual observations (e.g. atari flavors) or at least more nuanced ones (perhaps MiniGrid).

---

> ### Author Response · Authors · 2022-08-02
> **Response to Reviewer**
>
> >Strengths And Weaknesses:
>
> > Strengths
>
> > The paper is well-written and well-motivated. The method and design decisions are explained clearly.
> The analysis of the size of the task-relevant embedding (dimension d) is interesting and original.
>
> >Weaknesses
>
> > The experiments are limited to environments with small states, where there is a clear distinction between task-relevant dimensions and dimensions which can be discarded. The claims would be strengthened if experiments were extended to environments with visual observations (e.g. atari flavors) or at least more nuanced ones (perhaps MiniGrid).
>
> We agree that the application of our method to higher-dimensional observation spaces would be the logical next step to broaden its applicability. We believe that if the high-dimensional unstructured observation space is parsed into a meaningful abstract embedding space, our method should work well. At this point, we have run initial experiments with larger observations, which yield promising results but are not yet applicable at larger scale. More specifically, we utilize a pre-trained vision transformer and fine-tune its final attention layer with a loss function based on the mutual information objective in equation 7. In general, we see this as both a limitation and as an interesting direction for future work, and addressed this more clearly in the revised conclusion.
>
> > Questions:
>
> > I wonder if there are additional baselines that could be used. For example, what happens if vanilla imitation learning is done on the task-relevant embedding? I expect this would work poorly since the policy does need some environment-specific information.
>
> Thank you for this suggestion. Unfortunately this baseline cannot be implemented in a straightforward way, as the learner’s state is generally not of the same dimension as the task-relevant embedding. We instead added a baseline that is of similar fashion (please see next answer).
>
> > Another baseline could be some form of oracle in which the task-relevant dimensions are hand-picked or make use of the reward in some way.
>
> We added an additional oracle baseline in appendix 7.3.3. This baseline assumes that an oracle is used to find the learner’s state dimensions that match the task-relevant embedding of the expert state, while the exact order of the state dimensions is unknown. We then run imitation learning directly on the task-relevant embedding, i.e. omitting the learner encoder g. We find that, as hypothesised by the reviewer, this baseline performs poorly.
>
> > Limitations:
>
> > I'd like to see a discussion of limitations, perhaps addressing how the method might extend to more complex observations/environments.
>
> Thank you for this remark. We now addressed this more clearly in the conclusion.

---

### Author Response · Authors · 2022-08-02
**Response to all Reviewers**

We would like to thank all reviewers for their thorough and fair evaluations, and the valuable feedback. Below, please find the in-line answers to each of the reviews.

We further revised the manuscript accordingly and marked important changes in red (old version) and blue (new proposed version).

Three new experiments/ baselines requested by the reviewers can be found in the revised appendix, with results shown in Figures 8, 9 and 11.

Please let us know if any aspects remain unclear.

Best wishes,

The authors

---

### Author Response · Authors · 2022-08-08
**Discussion period ends tomorrow**

Dear reviewer 2, thank you for your feedback on our response. We likewise find that the suggested edits made our paper much better.

Dear reviewers 1 and 3, as the discussion period will end tomorrow, we just wanted to ask whether there are still any open questions?

Best wishes,
The authors.

---

### Meta-Review · Area_Chair_3ocr · 2022-08-24

**Recommendation:** Accept
**Confidence:** Certain

**Metareview:**

All three reviewers have elected to accept the paper, with two weak accepts and one accept. The reviews were thorough and demonstrated an understanding of the paper, and the authors have addressed many of the suggested edits.
I find figure 2 of the paper (comparison to XIRL on XMagical benchmark) compelling.
Recommendation: accept.

**Award:**

No

---

### Decision · Program_Chairs · 2022-09-14

Accept